# A Stochastic Deterioration Process Based Approach for Micro Switches Remaining Useful Life Estimation

**Bangcheng Zhang [1,\*], Yubo Shao [1] , Zhenchen Chang [2], Zhongbo Sun [1,3] and Yuankun Sui [4]**

1   School of Mechatronic Engineering, Changchun University of Technology, Changchun 130012, China; 13844989927@163.com (Y.S.); zhongbosun2012@163.com (Z.S.)
2   Crrc Changchun Rail Way Vehicles Co., Ltd., Changchun 130012, China; changzhenchen@cccar.com.cn
3   Key Laboratory of Bionic Engineering of Ministry of Education, Jilin University, Changchun 130025, China
4   COSMA Automotive (Shanghai) CO., LTD., Changchun 130000, China; 15662892233@163.com
*   Correspondence: zhangbangcheng@ccut.edu.cn; Tel.: +86-13331755427

**Abstract:** Real-time prediction of remaining useful life (RUL) is one of the most essential works in prognostics and health management (PHM) of the micro-switches. In this paper, a linear degradation model based on an inverse Kalman filter to imitate the stochastic deterioration process is proposed. First, Bayesian posterior estimation and expectation maximization (EM) algorithm are used to estimate the stochastic parameters. Second, an inverse Kalman filter is delivered to solve the errors in the initial parameters. In order to improve the accuracy of estimating nonlinear data, the strong tracking filtering (STF) method is used on the basis of Bayesian updating Third, the effectiveness of the proposed approach is validated on an experimental data relating to micro-switches for the rail vehicle. Additionally, it proposes another two methods for comparison to illustrate the effectiveness of the method with an inverse Kalman filter in this paper. In conclusion, a linear degradation model based on an inverse Kalman filter shall deal with errors in RUL estimation of the micro-switches excellently.

**Keywords:** micro-switches; remaining useful life; linear degradation model; inverse Kalman filter

---

## 1. Introduction

Nowadays, micro electro mechanical systems (MEMS) devices are used in various fields, such as automotive, biomedical, aerospace, and communication technologies [1]. They play an indispensable role in functioning and protection of the entire system [2]. As one of the components, micro-switches are affected by different working cycles and unavoidable external factors, such as changes in temperature and humidity, resulting in different degrees of residual life [3].

However, micro-switches reliability has attracted little attention, whose failure may cause significant downtime, as well as safety implications. Specifically, they are important parts of rail vehicle systems, and whether they are damaged is related to the operation of the entire system or even the safety of the passengers. Due to the significance of such aspect, several research works dealing with the reliability of micro-switches and other electronic components have been published, such as references [4–9]. Nevertheless, traditional approaches to estimate remaining useful life (RUL) have failed because of the reliance on average accumulated historical field data [10]. Reliability is estimated without taking into account the specific utilization of each component, such as working environment and using frequency. However, in practice, the lifetime should be different from one to another depending on how and where it is used. As a result, test duration and cost have become a huge challenge for traditional approaches. Real-time monitoring of the RUL of micro-switches and provide a convenient for timely maintenance decisions, is one of the important ways to improve its reliability [11,12].

The topic of the real-time prediction of RUL for electronic devices is one of the most active areas in prognostics and health management (PHM) research today. Considered with the stochastic characteristics of RUL in stochastic dynamic processes under actual working conditions, the data-driven RUL prediction was studied in the early 1980s, Derman et al. [13] confirmed the importance of life distribution in extending the life of equipment. This type of method is the most typical traditional life prediction method. The statistical analysis of life data determines the probability distribution of equipment life. However, the equipment such as micro-switches, owned high reliability and long-life features and it failed to obtain sufficient time to failure data in the short term, which made it difficult to obtain satisfactory prediction results for traditional life analysis methods based on life data. In recent years, there has been an increasing interest in the establishment of the real-time life prediction model by using monitoring data and calculating the probability distribution in the use of statistical methods. A large amount of literature has been delivered, Wang et al. [14] summarized the commonly used assumptions applying a random coefficient regression model and proposed a method for determining the failure threshold by optimization. Furthermore, Gebraeel et al. [15] proposed a logarithmic linearized exponential-like random coefficient regression model. The model utilized historical degradation data from similar devices and incorporated real-time monitoring data from historical data of service equipment through the Bayesian updating mechanism to update the remaining life distribution. Si et al. [16] summarized the experience of predecessors and provided an effective theory and method for establishing stochastic degradation models and studying RUL prediction problems. Especially, the algorithm put forward by Wang et al. [17] is widely used but there still exists some unsolved problems, such as, it is not sensitive to the initial real-time monitoring degradation data due to the objects of micro-switches.

In this paper, a linear degradation model based on an inverse Kalman filter to imitate the stochastic deterioration process was proposed. In addition, it referred to others about state monitoring methods, such as the extended Kalman filter (EKF) applied in the estimation of the position of the intake valve of the engine, and a theoretical basis, which was built up for the algorithm proposed in this paper [18,19]. Although Wang et al. [17] benefits were well proved, the algorithm was not sensitive to the initial real-time monitoring degradation data in solving the RUL estimation. Based on the Kalman filter, it was the important measure to the real-time condition monitoring of RUL, this paper proposed an inverse Kalman filter. Furthermore, Bayesian updating method and expectation maximization (EM) algorithm were used to estimate the RUL. Finally, the strong tracking filtering (STF) method was used to enhance the robustness. In order to verify the validity of the method, the S826 rail vehicle micro-switches were chosen as the research object. Thus, the real-time prediction of RUL for micro-switches were necessary. The data-driven method to solve the prediction of this kind of system could provide a feasible way for optimization of problems [20].

The remainder of this paper is organized as follows. Section 2 constructs a general stochastic process-based degradation model and then presents a degradation path-dependent approach for adaptive RUL estimation via real-time condition monitoring data. It discusses how to estimate initial parameters by using an inverse Kalman filter and illustrates a Bayes technique improved by the STF method, which can update the system parameters more accurately in real time. In Section 3, the test rig is designed to obtain performance degradation data for micro-switches. Section 4 provides several simulations and a case study to illustrate the application and usefulness of the developed approach. Section 5 concludes the paper.

Notations used in this paper.

| Notations | Explanations |
|---|---|
| $X_{0:k}$ | The degradation detection data $X_{0:k} = \{x_0, x_1, x_2, \ldots, x_k\}$. |
| $\theta$ | The drift parameter reflects the degradation rate of the equipment. |
| $\mu_{\theta,k}, \sigma^2_{\theta,k}$ | The updated hyper-parameters by the Bayesian posterior estimation. |
| $\Theta, \hat{\Theta}_k$ | $\Theta$ stands for the unknown parameters set are not updated by Bayesian estimation, $\Theta = [\sigma^2, \mu_0, \sigma^2_0]$, and $\hat{\Theta}_k$ denotes the updated results by the EM algorithm, $\hat{\Theta}_k = [\hat{\sigma}^2_k, \hat{\mu}_{0,k}, \hat{\sigma}^2_{0,k}]$. |
| $\nu(t_k)$ | The fading factor. |
| $P_{k|k}$ | The updated estimation variance by the STF technique. |
| $X'_{0:k}$ | $X_{0:k}$ is flipped as $X'_{0:k} = \{x'_0, x'_1, x'_2, \ldots, x'_k\}$, where $x'_0 = x_k, x'_1 = x_{k-1}, \cdots, x'_k = x_0$. |

## 2. Prognostic Approach

Under normal circumstances, most studies require multiple similar historical monitoring data to estimate parameters when they estimate the remaining useful life [21,22]. However, the micro-switches of the rail vehicle have high reliability and a long service life. For this type of component, even the accelerated life test requires considerable time and cost. Therefore, it is not feasible to estimate parameters with a large amount of historical data. Moreover, the running environment of each railway vehicle is different, and it is inaccurate to estimate the residual life with fixed parameters.

In this section, considering individual differences in micro-switches, we use real-time monitoring data to estimate individual systems. As the monitoring data is acquired, the system parameters will be updated adaptively, and an accurate prediction result can be obtained without similar historical data. The fundamental principle is that enough online monitoring data is used to complement the lack of similar historical data. A linear stochastic degenerate system is described in Figure 1.

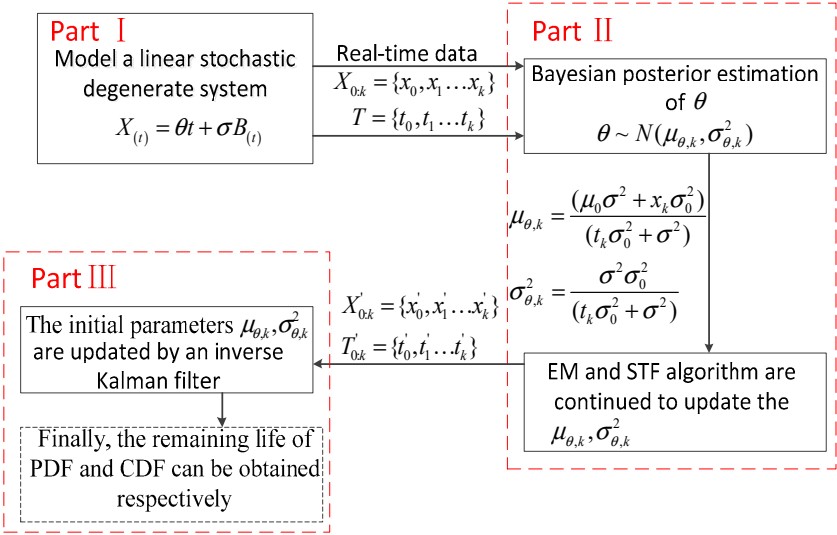

**Figure 1.** The flow chart of a linear stochastic degenerate system. EM: expectation maximization; STF: strong tracking filtering; PDF: probability density function; CDF: cumulative distribution function.

### 2.1. Modeling of Linear Stochastic Degenerate Systems

The linear degradation model is typically used for modelling degradation processes where the degradation rate is approximately a constant [17,23]. Moreover, the linear model with adaptive updating algorithm used in the literature [17] has a good estimate of the exponential type of data. In this paper, we consider the same linear degradation model based on a Wiener process as follows:

$$X(t) = \theta t + \sigma B(t) \tag{1}$$

where $X(t)$ is the degradation detection data at time $t$. The initial state is shown as $t = 0, X(0) = 0$. To be consistent with existing studies, the initial degradation was 0, which can be obtained by translation transformation of the data. And $\theta$ was the drift parameter, which means the degradation rate of the system, $\sigma$ is the diffusion parameter. $B(t)$ denotes the standard Brownian motion, which represents the stochastic dynamics of the degradation process. The degradation detection data is described as $X_{0:k} = \{x_0, x_1, x_2, \ldots, x_k\}$.

The reason for using linear system equations is not only its universality, but also it is the convenience for calculation. Another important reason is that the life prediction problem can be understood as predicting the future trend of a random curve. According to the Euler method, given a starting point, if we can predict the tangent curve at any point, from the starting point, we can calculate the predictive value step by step, the approximate curve can be obtained. In this paper, the drift parameter $\theta$ denotes the tangent curve and we use a reasonable method to adjust $\theta$ in real time. From the Equation (1), each step of the degradation data can be expressed as:

$$X(t_k - t_{k-1}) = \theta(t_k - t_{k-1}) + \sigma B(t_k - t_{k-1}) \tag{2}$$

where $X(t_k - t_{k-1}) \sim N\big(\theta(t_k - t_{k-1}), \sigma^2(t_k - t_{k-1})\big)$.

It can also be described in a nonlinear model, which will greatly increase computation time and not improve performance significantly. In addition, we apply Bayesian updating and the EM algorithm to update the system parameters, and the STF is used to improve the robustness of the model parameter mismatch. The above methods can guarantee the accuracy of parameters. The detailed description will be given below.

Assumption: In the model of $X(t) = \theta t + \sigma B(t)$, $\theta$ is assumed to be a random parameter, indicating individual differences. $\sigma$ is assumed to be the deterministic parameter as a constant.

## 2.2. Bayesian Posterior Estimation of Stochastic Parameters

It is noticed that the key parameter for determining the degradation state is $\theta$ in Equation (1). $\theta$ is a random parameter and will be updated with the data obtained at the current moment $t_k$. $\theta$ is distributed as $\theta \sim N\left(\mu_{\theta,k}, \sigma_{\theta,k}^2\right)$, such parameter distributions are consistent with existing methods [17,23].

In order to estimate hyper-parameters $\mu_{\theta,k}, \sigma_{\theta,k}^2$ in the random parameter $\theta$, the Bayesian posterior estimation is used in this paper.

Firstly, it is assumed that the prior distribution of $\theta$ is $N\left(\mu_0, \sigma_0^2\right)$. Then the chain recursion was incorporated into the calculation. The posterior distribution of $P(\theta|X_{0:k})$ can be expressed as [17]:

$$p(\theta|X_{0:k}) = \frac{1}{\sigma_{\theta,k}\sqrt{2\pi}} \exp\left\{-\frac{(\theta - \mu_{\theta,k})^2}{2\sigma_{\theta,k}^2}\right\} \tag{3}$$

Where

$$\mu_{\theta,k} = \frac{\left(\mu_0\sigma^2 + x_k\sigma_0^2\right)}{\left(t_k\sigma_0^2 + \sigma^2\right)}, \sigma_{\theta,k}^2 = \frac{\sigma^2\sigma_0^2}{\left(t_k\sigma_0^2 + \sigma^2\right)} \tag{4}$$

It can be found in Equation (4) that the posterior estimate of the random parameter $\theta$ can be updated after the new monitoring data.

## 2.3. Estimation of Unknown Parameters Based on EM Algorithm

As seen in the previous section, unknown parameters $\Theta = [\sigma^2, \mu_0, \sigma_0^2]$ are not updated by the Bayesian estimation. The reason for using the EM algorithm instead of the maximum likelihood estimation algorithm is that the unknown parameter $\Theta$ contains the hidden variable $\theta$, which cannot be directly estimated by the maximum likelihood estimation. We need to approximate the maximum likelihood estimation of parameters by maximizing the joint likelihood function $p(X_{0:k}, \theta|\Theta_k)$. In order

to reflect the updated characteristics of $\Theta$ over time, we use the EM algorithm to estimate $\Theta$ through monitoring data $X_{0:k}$, its update results are expressed as $\hat{\Theta}_k = [\hat{\sigma}_k^2, \hat{\mu}_{0,k}, \hat{\sigma}_{0,k}^2]$ [17].

First, complete logarithm likelihood function $\ln p(X_{0:k}, \theta|\Theta_k)$ is calculated as:

$$
\begin{aligned}
\ln p(X_{0:k}, \theta|\Theta_k) &= \ln p(X_{0:k}|\theta, \Theta_k) + \ln p(\theta|\Theta_k) \\
&= -\frac{k+1}{2}\ln 2\pi - \frac{1}{2}\sum_{j=1}^k \ln(t_j - t_{j-1}) - \frac{k}{2}\ln\sigma_k^2 - \\
&\quad \sum_{j=1}^k \frac{\left(x_j - x_{j-1} - \theta(t_j - t_{j-1})\right)^2}{2\sigma_k^2(t_j - t_{j-1})} - \frac{1}{2}\ln\sigma_{0,k}^2 - \frac{(\theta - \mu_{0,k})^2}{2\sigma_{0,k}^2}
\end{aligned}
\tag{5}
$$

E-step: Calculate the expected value $\ell\left(\Theta_k \middle| \hat{\Theta}_k^{(i)}\right)$ of $\ln p(X_{0:k}, \theta|\Theta_k)$ which is about $p(\theta|X_{0:k}, \Theta_k^{(i)})$

$$
\begin{aligned}
\ell\left(\Theta_k \middle| \hat{\Theta}_k^{(i)}\right) &= \mathrm{E}_{\theta|X_{0:k}, \hat{\Theta}_k^{(i)}}\{\ln p(X_{0:k}, \theta|\Theta_k)\} \\
&= -\frac{k+1}{2}\ln 2\pi - \frac{1}{2}\sum_{j=1}^k \ln(t_j - t_{j-1}) - \frac{k}{2}\ln\sigma_k^2 \\
&\quad - \sum_{j=1}^k \frac{(x_j - x_{j-1})^2 - 2\mu_{\theta,k}(t_j - t_{j-1})(x_j - x_{j-1}) + (t_j - t_{j-1})^2(\mu_{\theta,k}^2 + \sigma_{\theta,k}^2)}{2\sigma_k^2(t_j - t_{j-1})} - \frac{1}{2}\ln\sigma_{0,k}^2 - \frac{\mu_{\theta,k}^2 + \sigma_{\theta,k}^2 - 2\mu_{\theta,k}\mu_{0,k} + \mu_{0,k}^2}{2\sigma_{0,k}^2}
\end{aligned}
\tag{6}
$$

M-step: Fixed parameter $\theta$, and take the maximum value of $\Theta$. $\partial\ell\left(\Theta_k \middle| \hat{\Theta}_k^{(i)}\right)/\partial\Theta_k = 0$, $\hat{\Theta}_k^{(i+1)}$ can be expressed as follows:

$$
\begin{aligned}
\hat{\sigma}_k^{2(i+1)} &= \frac{1}{k}\sum_{j=1}^k \frac{(x_j - x_{j-1})^2 - 2\mu_{\theta,k}(t_j - t_{j-1})(x_j - x_{j-1}) + (t_j - t_{j-1})^2(\mu_{\theta,k}^2 + \sigma_{\theta,k}^2)}{(t_j - t_{j-1})}, \\
\hat{\mu}_{0,k}^{(i+1)} &= \mu_{\theta,k}, \quad \hat{\sigma}_{0,k}^{2\,(i+1)} = \sigma_{\theta,k}^2
\end{aligned}
\tag{7}
$$

Moreover, the updated results in Equation (7) required only one step to compute the maximum value, which have been given proof by the literature [17]. One step to solve the maximum value greatly reduces the computing time and has a strong practical value.

From the results of Equation (7), the main updated parameter of the EM is $\sigma^2$. The other two parameters $\mu_{\theta,k}, \sigma_{\theta,k}^2$ are also updated. The initial parameters in the Bayesian estimation are improved after Bayesian updating in the next step.

## 2.4. Adding Fading Factor Based on the STF to Enhance Robustness

In this section, we discuss how to adjust parameters in time and guarantee the accuracy of estimation when the model parameters and real-time data are mismatched.

It can be proved from Equations (4) and (7):

$$
\begin{aligned}
\sigma_{\theta,k-1}^2 &= \sigma_0^2, \\
\sigma_{\theta,k}^2 &= \frac{\sigma^2\sigma_{\theta,k-1}^2}{t_k\sigma_{\theta,k-1}^2 + \sigma^2} = \sigma_{\theta,k-1}^2 \cdot \frac{\sigma^2}{t_k\sigma_{\theta,k-1}^2 + \sigma^2} < \sigma_{\theta,k-1}^2
\end{aligned}
\tag{8}
$$

This means that the value of the parameter $\sigma_{\theta,k}^2$ will gradually decrease as the algorithm is updated. That is, the uncertainty of the real value is decreasing. However, when enough data is available, Equation (8) can be expressed as:

$$
\lim_{k\to\infty}\sigma_{\theta,k}^2 = \lim_{k\to\infty}\frac{\sigma^2}{t_k + \frac{\sigma^2}{\sigma_{\theta,k-1}^2}} = \lim_{k\to\infty}\frac{1}{\frac{t_k}{\sigma^2} + \frac{1}{\sigma_{\theta,k-1}^2}} = 0
\tag{9}
$$

It is easy to prove that $\sigma_{\theta,k}^2$ will approach to 0 when $t_k \to \infty$. And because $\sigma_{\theta,k-1}^2$ in the Equation (9) is also monotonically decreasing, hyper-parameter $\sigma_{\theta,k}^2$ will decay faster. When $\sigma_{\theta,k}^2 \to 0$, it can be seen that:

$$\lim_{\sigma_{\theta,k}^2 \to 0} \mu_{\theta,k} = \lim_{\sigma_{\theta,k}^2 \to 0} \frac{\left(\mu_{\theta,k-1}\sigma^2 + x_k\sigma_{\theta,k}^2\right)}{\left(t_k\sigma_{\theta,k}^2 + \sigma^2\right)} = \mu_{\theta,k-1} \tag{10}$$

$\mu_{\theta,k}$ will not change as new data is acquired from Equation (10), that is, the stochastic degradation parameter $\theta$ will not change with the acquisition of new data. Wang et al. [17] directly uses Bayesian updating and EM algorithm to estimate unknown parameters $\Theta$. When the degraded data is stationary, it can be well estimated. However, when parameters are convergent, it will not obtain good estimation results if the newly acquired data is different from the model parameter.

The reason for the parameter no longer being updated is $\sigma_{\theta,k}^2 \to 0$. Considering that the Kalman filter algorithm is obtained in the Bayesian framework, it has some similarities with the Bayesian updating. The Kalman filter results in the fact that the *K* value tends to the minimum, thus that it is no longer sensitive to the prediction error. The STF solves the problem of mutational degradation data on the basis of the Kalman filter [24]. It is not practical that $\sigma_{\theta,k}^2$ is rapidly approaching 0, inspired by reference [24], we added the prediction error *Q* in each step update to ensure the update capability of the algorithm, then we added a fading factor to adjust the prediction variance in real time, thus that $\sigma_{\theta,k}^2$ could be sensitive to the prediction error. The specific calculation steps are as follows:

Step 1: Establish system equation

$$\mu_{\theta,k} = \mu_{\theta,k-1} + \eta \ , \quad \eta \sim N\left(0, \sigma_{\theta,k-1}^2\right) \tag{11}$$

$$\begin{aligned} x_k - x_{k-1} &= \mu_{\theta,k}(t_k - t_{k-1}) + \sigma_k\varepsilon'(t_k), \\ \varepsilon'(t_k) &\sim N\left(0, \sigma_k^2(t_k - t_{k-1})\right) \end{aligned} \tag{12}$$

Step 2: Set initial parameters $\mu_0, \sigma_0, \alpha, \rho$

Step 3: Calculated fading factor $v(t_k)$

$$\gamma(t_k) = x_k - x_{k-1} - \mu_{\theta,k}(t_k - t_{k-1}) \tag{13}$$

$$V_0(t_k) = \begin{cases} \gamma^2(t_1), k = 1 \\ \frac{\rho V_0(t_{k-1}) + \gamma^2(t_k)}{1+\rho}, k > 1 \end{cases} \tag{14}$$

$$B(t_k) = V_0(t_k) - \sigma_{\theta,k}^2(t_k - t_{k-1})^2 - \alpha\sigma_k^2(t_k - t_{k-1}) \tag{15}$$

$$C(t_k) = P_{k-1|k-1}(t_k - t_{k-1})^2, v_0 = \frac{B(t_k)}{C(t_k)} \tag{16}$$

$$v(t_k) = \begin{cases} v_0, v_0 \geq 1 \\ 1, v_0 < 1 \end{cases} \tag{17}$$

Step 4: Status updates

$$P_{k-1|k} = v(t_k)P_{k-1|k-1} + \sigma_{\theta,k}^2 \tag{18}$$

$$K_k = \frac{P_{k|k-1}(t_k - t_{k-1})}{(t_k - t_{k-1})^2 P_{k|k-1} + \sigma_k^2(t_k - t_{k-1})} \tag{19}$$

Model parameter updating:

$$\hat{\mu}_{\theta,k} = \mu_{\theta,k} + K_k \cdot \gamma(t_k) \tag{20}$$

Estimation variance updating:

$$P_{k|k} = (1 - K_k \cdot t_k)P_{k|k-1} \tag{21}$$

Usually, the forget factor is set to $\rho = 0.95$, the softening coefficient is $\alpha = 1.1$, and the prediction error is $Q = 0.5$.

## 2.5. Estimating Initial Parameters by an Inverse Kalman Filter

Although the algorithm by Wang et al. [17] has self-adaption update capability, even if the initial parameters are not set correctly, it will approach the accurate value as the new data is acquired. However, a practical problem is that the initial parameters are difficult to determine without a large amount of historical information, and it is also unknown where monitoring data starts from the whole life cycle. A progressively updating algorithm of parameter $\theta$ is proposed by Wang et al. [17]. If the initial parameter is set to be more inconsistent with the actual, such errors will affect the subsequent estimation process, and its convergence rate will be slow, resulting in that a large amount of monitoring data is needed to complete the parameter convergence. Even more, the algorithm has not completed convergence, experimental object has been damaged and the inaccurate life prediction results are obtained in its life cycle. To this end, this section will discuss an inverse Kalman filter to update the initial parameters in real time. The initial parameters are updated at the source, thus that the convergence speed of the estimation is accelerated.

The reason for this application of an inverse Kalman filter is that the drift parameter $\theta$ is unobservable, where the nonlinear form adopted is an unknown problem. Because the degradation is more stable in the early phase, when the sampling interval is not long, it is reasonable for $\mu_{\theta,k}$ to obey stochastic Gauss distribution around $\mu_{\theta,k-1}$ based on large sample statistical theory [25].

At the same time, nonlinear modeling methods (such as exponential models) can be used, and model parameters can be updated with the Bayesian updating and EM. After model parameters are determined, the initial parameters will be estimated by an inverse Kalman filter. In this way, the algorithm will be very complex and computation time will be greatly increased. More importantly, although the initial phase of degradation is relatively stable, it does not necessarily satisfy the overall degradation model. For example, the initial degradation process data in this paper does not agree with the overall degradation trend. That is, the model that conforms to the overall data may not necessarily satisfy the initial degradation process, it is related to the time point of when the monitoring begins. Therefore, it is difficult to estimate the exact initial parameters by using the determined nonlinear model.

It is often the case with actual degradation data: The initial degradation phase is stable, and then the volatility of data becomes more pronounced until the system fails, such as the data from this paper and reference [26]. Therefore, it is more and more difficult to update the initial parameters by normal methods. Initial parameters are difficult to converge, even if the initial parameters are convergent, they are not the exact values. If the initial parameters are updated via recursively in reverse, it will be found that degradation data will be smooth gradually, updating the true value of the initial parameters $\mu_{\theta,k}, \sigma^2_{\theta,k}$ that are expected to be improved, and converge to the exact value.

For the actual operation of the system, the monitoring data corresponding to the current time $t_k$ is $x_k$, and the observation data are $X_{0:k} = \{x_0, x_1, x_2, \ldots, x_k\}$. In order to show the Kalman updated process is more intuitive, the order of the elements are flipped as $X'_{0:k} = \{x'_0, x'_1, x'_2, \ldots, x'_k\}$, where $x'_0 = x_k, x'_1 = x_{k-1}, \cdots, x'_k = x_0$. $T_{0:k}$ has been rewritten as $T'_{0:k} = \{t'_0, t'_1, t'_2, \ldots, t'_k\}$ by the same method.

According to the conventional Kalman filter [25,27], all monitoring data of the system is recursively incorporated into parameter $\theta$ by an inverse Kalman filter technology. The specific calculation process is as follows:

Step 1: Establish system equation

$$\mu_k = \mu_{k-1} + \eta' \tag{22}$$

$$x'_k - x'_{k-1} = (t'_k - t'_{k-1}) \cdot \mu_k + \sigma \varepsilon_k \tag{23}$$

where $\eta \sim N(0, Q)$, $\sigma \varepsilon_k \sim N(0, \sigma^2(t'_k - t'_{k-1}))$, $Q$ is the forecast variance.

Step 2: Time updating

$$\hat{\mu}_k^- = \hat{\mu}_{k-1} \tag{24}$$

$$P_{k|k-1} = P_{k-1|k-1} + Q \tag{25}$$

where $\hat{\mu}_0 = -\mu_0$, $P_{0|0} = \sigma_0^2$.

Step 3: Status updating

$$K_k = \frac{P_{k|k-1} \cdot \left(t_k' - t_{k-1}'\right)}{\left(t_k' - t_{k-1}'\right)^2 \cdot P_{k|k-1} + \sigma^2 \left(t_k' - t_{k-1}'\right)} \tag{26}$$

$$\hat{\mu}_k = \hat{\mu}_k^- + K_k \left( \left(x_k' - x_{k-1}'\right) - \left(t_k' - t_{k-1}'\right) \hat{\mu}_k^- \right) \tag{27}$$

$$P_{k|k} = \left(1 - K_k \cdot \left(t_k' - t_{k-1}'\right)\right) P_{k|k-1} \tag{28}$$

where $R$ is the system error, $R = \sigma^2 \left(t_1' - t_0'\right)$.

Step 4: Update results of the initial parameter

$$\mu_{\theta,k} = -\hat{\mu}_k \tag{29}$$

$$\sigma_{\theta,k}^2 = P_{k|k} \tag{30}$$

The parameters of an inverse Kalman filter are set as $R = 0.025$ $Q = 0.50$, which can make the algorithm more dependent on system measurements.

**Remark 1.** *For an inverse Kalman filter, the difference from the conventional Kalman filter [25,27] is that the order of the elements $X_{0:k}$ are flipped as $X_{0:k}' = \left\{x_0', x_1', x_2', \ldots, x_k'\right\}$, it is accounted for $x_0' = x_k$, $x_1' = x_{k-1}, \cdots, x_k' = x_0$, $x_k$ is the last monitoring data at the current time $t_k$, $x_0$ is the initial monitoring data at the current time $t_0$. The flipped data as $X_{0:k}' = \left\{x_0', x_1', x_2', \ldots, x_k'\right\}$ guarantees the iteratively updated forward of the estimation from the last monitoring point when it starts to filter.*

**Remark 2.** *The consequence of applied an inverse Kalman filter reveals that it can be obtained by the optimal estimation $\mu_{\theta,k}$.*

**Remark 3.** *It is obvious that the steps according to an inverse Kalman filter are similar with the conventional Kalman filter. However, a practical problem is that the initial parameters are difficult to determine without a large amount of historical information, thus they are indeterminate and often set with errors. If we take advantage of the conventional Kalman filter to solve the errors belonging to the initial parameters, it will not be sensitive to them. In contrast, the last parameters are fixed and reliable relatively accounts for a number of historical information. The advantage of an inverse Kalman filter is that, when we start to filter from the last monitoring data $x_k$, the accuracy of the filtering is improved for the initial monitoring data especially.*

*2.6. Expression of Remaining Useful Life*

Based on the concept of stochastic process lead time, when the failure threshold $\omega$ is reached for the first time, the system life is considered to be terminated. Based on the observed data $X_{0:k} = \{x_0, x_1, x_2, \ldots, x_k\}$, the RUL $L_k$ of the system at the moment $t_k$ is defined as:

$$L_k = \inf\{l_k : X(l_k + t_k) \geq \omega | X_{0:k}\} \tag{31}$$

After getting the new data and corresponding updated parameters $\Theta = [\sigma^2, \mu_{\theta,k}, \sigma^2_{\theta,k}]$, According to the literature [17], the remaining life of PDF (probability density function) and CDF (cumulative distribution function) can be obtained respectively:

$$f_{L_k|X_{0:k}}(l_k|X_{0:k}) = \frac{\omega - x_k}{\sqrt{2\pi l_k^3 \left(\sigma^2_{\theta,k} l_k + \sigma^2\right)}} \exp\left\{-\frac{(\omega - x_k - \mu_{\theta,k} l_k)^2}{2l_k\left(\sigma^2_{\theta,k} l_k + \sigma^2\right)}\right\} \tag{32}$$

$$F_{L_k|X_{0:k}}(l_k|X_{0:k}) = 1 - \Phi\left(\frac{\omega - x_k - \mu_{\theta,k} l_k}{\sqrt{\sigma^2_{\theta,k} l_k^2 + \sigma^2 l_k}}\right) + \exp\left\{\frac{2\mu_{\theta,k}(\omega - x_k)}{\sigma^2} + \frac{2\sigma^2_{\theta,k}(\omega - x_k)^2}{\sigma^4}\right\} \times \left(-\frac{2\sigma^2_{\theta,k}(\omega - x_k)l_k + \sigma^2\left(\mu_{\theta,k} l_k + \omega - x_k\right)}{\sigma^2\sqrt{\left(\sigma^2_{\theta,k} l_k^2 + \sigma^2 l_k\right)}}\right) \tag{33}$$

Here, the life prediction method under the linear stochastic deterioration model has been completed. The following steps are concluded to estimate the RUL of the micro-switches, which belongs to the S826 rail vehicle:

Step 1: A linear degradation model based on a Wiener process is proposed: $X(t) = \theta t + \sigma B(t)$, the degradation detection data are described as $X_{0:k} = \{x_0, x_1, x_2, \ldots, x_k\}$.

Step 2: $\theta$ is a random parameter and will be updated with the data obtained at the current moment $t_k$. $\theta$ is distributed as $\theta \sim N\left(\mu_{\theta,k}, \sigma^2_{\theta,k}\right)$, in order to estimate the hyper-parameters $\mu_{\theta,k}, \sigma^2_{\theta,k}$ in the random parameter $\theta$, the Bayesian posterior estimation is used in this paper. Finally, calculate the expression of the hyper-parameters $\mu_{\theta,k}, \sigma^2_{\theta,k}$.

Step 3: Unfortunately, the unknown parameters $\Theta = [\sigma^2, \mu_0, \sigma^2_0]$ are not updated by Bayesian estimation. In order to reflect the updated characteristics of $\Theta$ over time, we use the EM algorithm to estimate $\Theta$ through the monitoring data $X_{0:k}$, and its update results are expressed as $\hat{\Theta}_k = [\hat{\sigma}^2_k, \hat{\mu}_{0,k}, \hat{\sigma}^2_{0,k}]$. The main updated parameter of the EM is $\sigma^2$, other two parameters $\mu_{\theta,k}, \sigma^2_{\theta,k}$ are also updated. Initial parameters-based Bayesian estimation are improved after Bayesian updating in the next step.

Step 4: When the parameters are convergent, it will not obtain good estimation results if the newly acquired data is different from the model parameter. The STF method solves the problem of mutational degradation data on the basis of the Kalman filter. We add prediction error $Q$ in each step update to ensure the update capability of the algorithm, then we add the fading factor to adjust the prediction variance in real time, thus that $\sigma^2_{\theta,k}$ can be sensitive to the prediction error.

Step 5: The drift parameter $\theta$ is unobservable, and the nonlinear form is adopted to an unknown problem. The initial parameter is set to be more inconsistent with the actual parameter, and such errors will affect the subsequent estimation process, and its convergence rate will be slower, resulting in a large amount of monitoring data being needed to complete the parameter convergence. As considered above, an inverse Kalman filter is proposed to update the initial parameters in real time, as well as updating at source, thus that the convergence speed of the estimation is accelerated.

## 3. Experimental Setup and Tests

In order to verify the effectiveness of our method applied to micro-switches, a test rig should be designed to record the real-time degradation data.

### 3.1. The Establishment of the Test Rig

Arcing is the main factor causing the micro-switches to fail [28]. Whenever the switch contacts are separated, the arc will be generated, and the contact voltage will continue to rise until failure [29].

When it starts to work, there exists two phenomenon in the process. One is increasing in contact voltage: For micro-switches, the arc generated on the contacts will be as high as 4000 K or more, making the contact material partially melted and sputtered. In the meanwhile, it generates complex physical and chemical processes. With the increase of the work cycle, there are thousands of these repeated effects and superposition, and the contact resistance gradually increases until the conduction capability is lost. The other is insulation performance reduction: When micro-switches are in operation,

due to the repeated action of the arc, melting, vaporization, and splashing of the electric shock material, the metal compound adheres to the surface of the insulating part near the contact. With the increase in the number of work, the attached crop will grow thicker until it connects with the insulated conductors. The working period of typical drive controller micro-switch is shown in Figure 2.

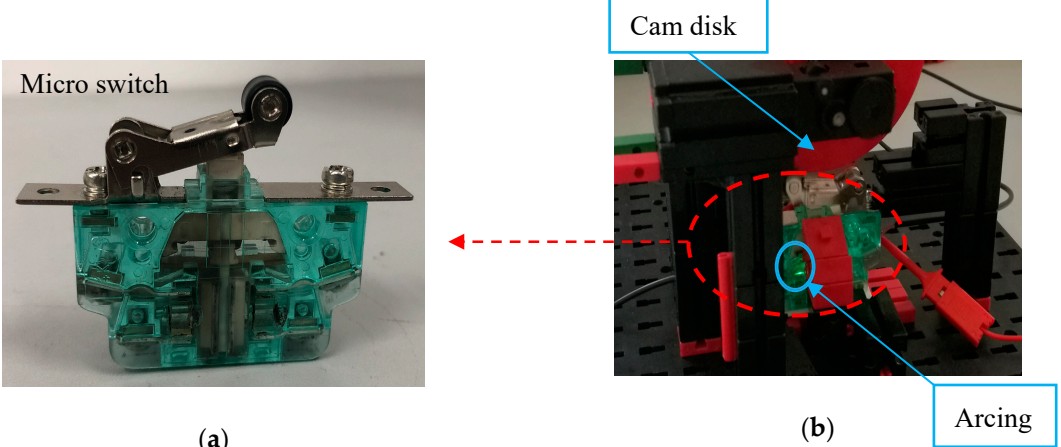

**Figure 2.** The working period of the typical drive controller micro-switch (S826).

Due to the actual working conditions of the micro-switches, some basic physical quantities were selected. Rated voltage is chosen as 110V direct current (DC), the rated current was chosen as 1A DC, the time constant was chosen as 15 ms and the operating frequency was chosen as 120r/min [30].

The test rig used in this experiment was identical with the one used by Zhang et.al. [30], designed to test the life of micro-switches showing in Figure 3.

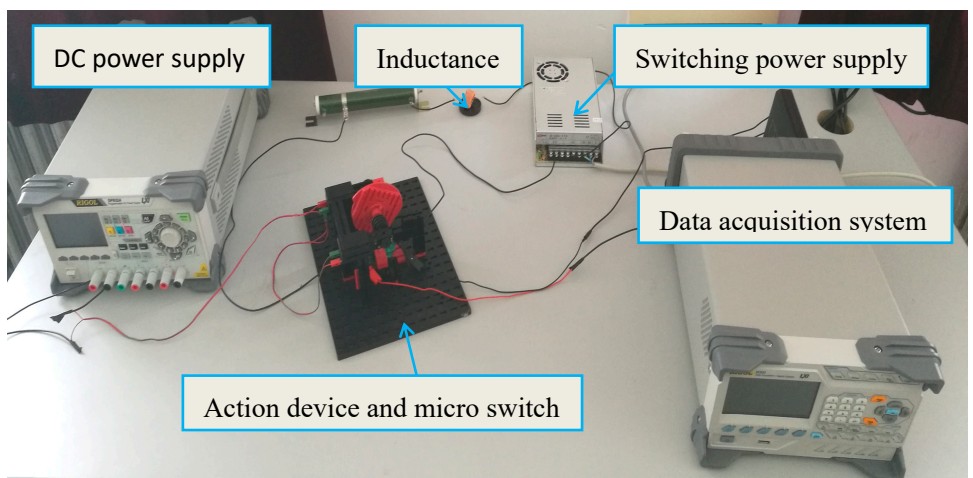

**Figure 3.** The test rig of micro-switches.

### 3.2. The Collection of Experimental Data

In this experiment, which is similar with the paper [30], 160,800 contact voltage data were collected. Then the micro-switches were failed, and the resistance remained constant in 5.34 MΩ, which is a normally open state, thus the micro-switch was determined to fail. The failure threshold was set to 1.80 V. On the basis of not losing the monitoring information, we processed the degradeded data in order to represent the degradation process of the whole data. The average value of the dynamic contact voltage drop of each 600 cycles was recorded until the end of life (Figure 4).

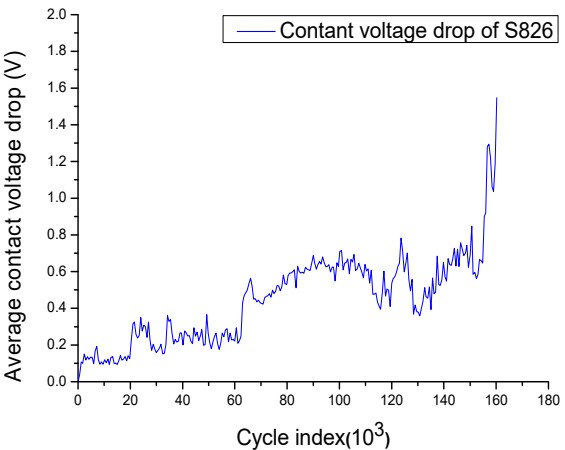

**Figure 4.** Dynamic contact voltage drop of S826.

From the degraded data, it can be seen that the initial phase of degradation was similar to the linear stochastic degradation process. However, the last 20 points are growing rapidly.

Results

Our approach in this paper was used to simulate the degradation path of the dynamic contact voltage drop, as shown in Figure 5. As seen from this approach, whether it is in the initial and final monitoring phase, our approach fits very well. As considered less previously, for initial parameters, we used an inverse Kalman filter for the initial data update. It can be seen that an inverse Kalman filter is still sensitive to the drift parameter $\theta$ in the case of less initial parameters. Furthermore, in order to enhance the robustness in the process of estimation, we added the fading factor based on the STF.

In order to show the superiority of this method, the initial parameters of the model were set as $\Theta_0 = [2, 0.001, 0.4]$, and the updated process of the parameter is shown in Figure 6. The results show that the accumulation of model parameters can converge quickly and can adjust slightly with the change of the degradation tendency.

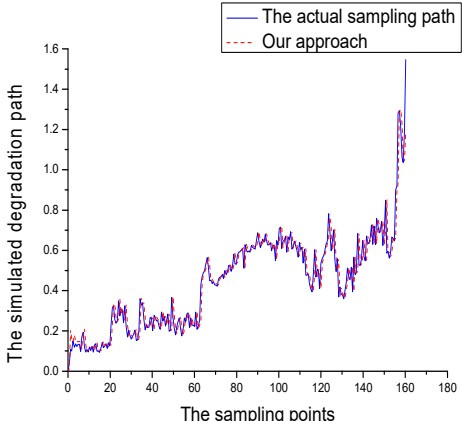

**Figure 5.** Our approach simulates the degradation path.

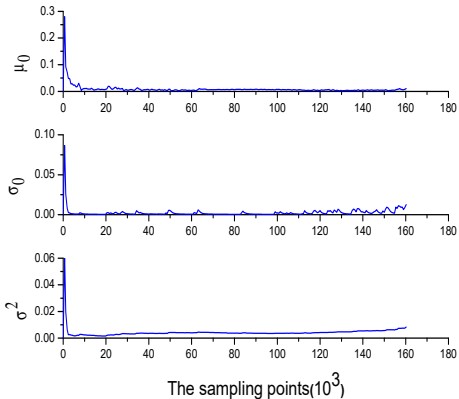

**Figure 6.** Adaptive estimation process of the model parameters.

## 4. Comparative Studies

In this section, we used the test data of the S826 micro-switches to illustrate the practicability of this research result, comparing it with the approaches, the Kalman filter instead of an inverse Kalman filter, and the algorithm of Wang et al. [17] in order to verify the superiority of our method.

For the last 80 sampling points, the method with the Kalman filter instead of an inverse Kalman filter was similar with our approach, thus Figure 7 compared the updated parameters obtained by Wang et al. [17] with the method proposed in this paper. For these two approaches, the unknown parameters were obtained by the combination of the Bayesian updating and the EM algorithm. The difference is that the fading factor was added in this paper, thus that the drift coefficient was more sensitive to the change of data. As can be seen from the diagram, the prediction error approaches zero in the approach by Wang et al. [17], resulting in $\mu_0$ being not sensitive to new data, when the micro-switch is about to fail, the degradation rate was obviously accelerated, and our approach could be adjusted better.

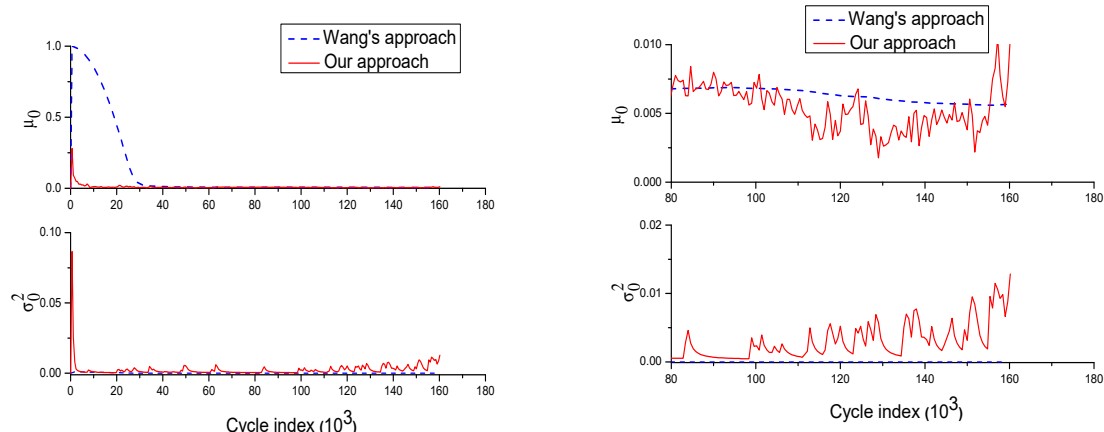

**Figure 7.** Compare the last 80 sampling points of the model parameters.

The literature [17] requires higher precision of initial parameter selection, and it is difficult to set an accurate initial parameter in practical applications, because there does not exist a large number of similar historical information and the accurate time to start data monitoring is unknown. In this paper, we selected a set of relatively inaccurate initial parameters in two methods $\Theta_0$ to verify the ability of our algorithm about updating the initial parameters. It can be seen from the comparison chart (Figure 8) that when the parameters are improperly set, the convergence speed of Wang et al. [17] is slow. Furthermore, the method proposed in this paper has faster convergence speed in the initial few sampling points than the method with Kalman filter instead of an inverse Kalman filter merely.

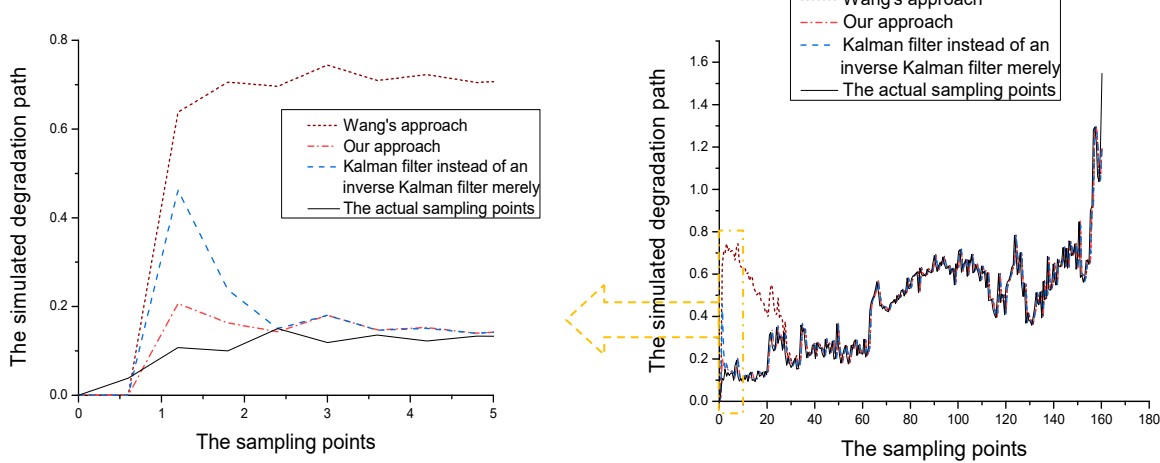

**Figure 8.** The comparison chart.

Figure 9 reflects the mean square error (MSE) values at different monitoring time points. In the initial phase, degradation data are less, the fluctuation of the method proposed by Wang et al. [17] is the largest and with the smallest fluctuation is our method. This means that the remaining life of the PDF of another two predictive models are sensitive to small changes, and if it is applied for a maintenance decision, it may result in two different monitoring points, which are completely different to the maintenance decisions which increases protection and maintenance costs in turn. As a conclusion, our approach has a higher prediction accuracy. Notice that Figure 9b shows an upward trend of MSE, it is mainly because when the life is about to terminate, the data have fluctuated greatly, and the error has been raised slightly.

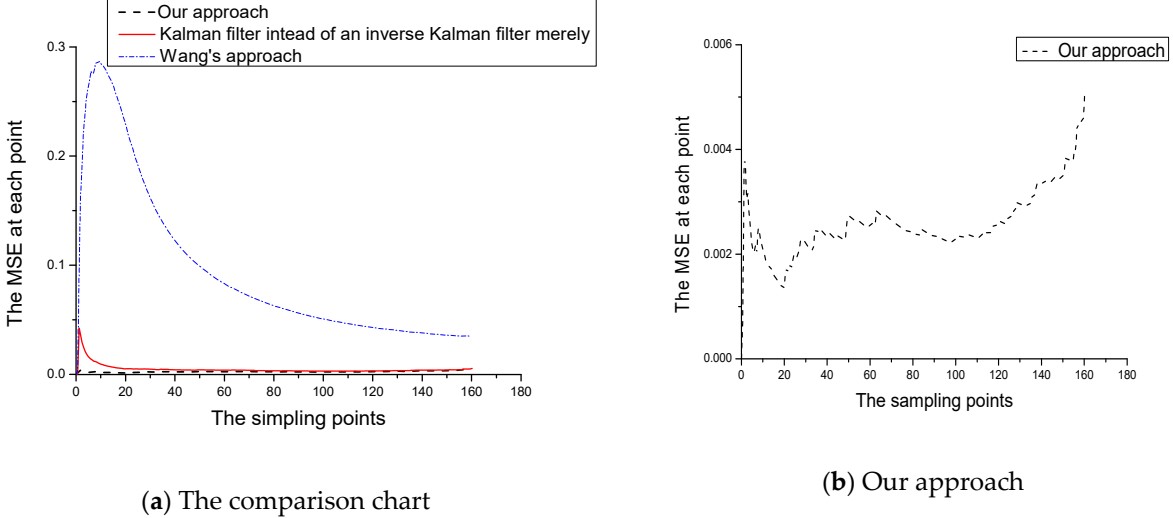

(**a**) The comparison chart　　　　　　　　　　　　　　　　(**b**) Our approach

**Figure 9.** Mean square error (MSE) of expectation for all monitoring points.

Figure 10 illustrates the approach proposed by Wang et al. [17], which compares with the one proposed by us regarding the estimation RUL at the last four sampling points. PDF becomes gently sharper and closer to the Z-axis by applying our approach. This means when more data are used to estimate parameters, the uncertainty of the remaining life is decreasing, which agrees with the facts.

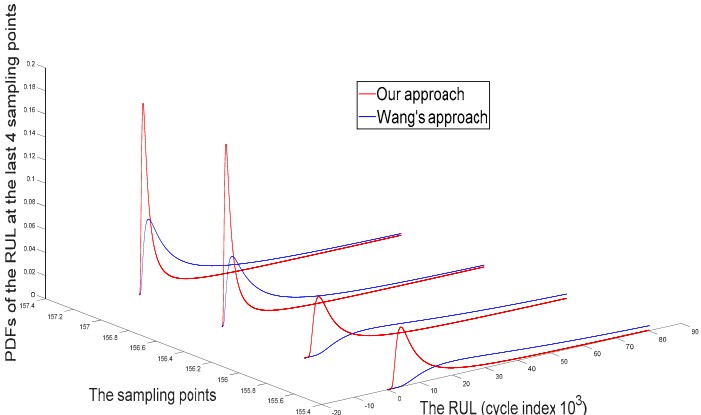

**Figure 10.** The PDF of the remaining useful life (RUL) at the last four different monitoring points.

**Remark 4.** *It can't be ignored to illustrate a superior method between the algorithm proposed by Zhang et al. [30] and us. Figure 11 reflects that our method fits the actual degradation path better than Zhang et al. [30], especially in the initial several sampling points. It is more obvious in Figure 12, where our PDF is closer to the z-axis than shown by Zhang et al. [30]. This means our approach is still very sensitive to the monitoring data in the final phase.*

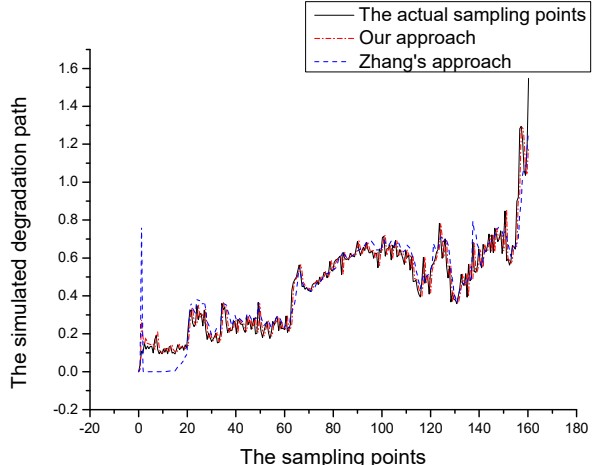

**Figure 11.** The comparison chart.

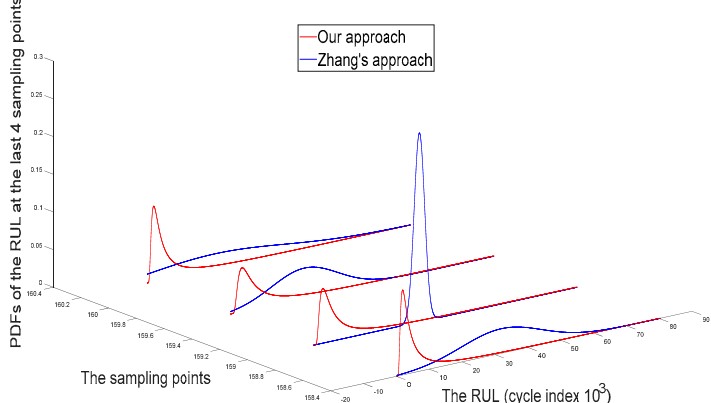

**Figure 12.** The PDF of RUL at the last four different monitoring points.

In conclusion, our approach is more sensitive and adjustable to degradation data of micro-switches than the one proposed by Zhang et al. [30].

*Discussion*

The most remarkable result that emerged from the data were our approach with an inverse Kalman filter that fitted the real degradation path excellently. Our results shared a number of similarities with findings according to the literature [17,25]. However, different from earlier findings, our approach dealt better with errors in the initial degradation phase. We put forward another two methods to compare and demonstrate our view. In addition, we also compared this with Zhang et al. [30] and got a satisfied result. It is easily seen in the previous study; our approach is the most sensitive to the actual sampling points. The results we have obtained will provide strong technical support for PHM, including micro-switches and even other electronic components of the rail vehicles. And it will be a solid basic study about nonlinear degradation path based on the electronic components in the future.

## 5. Conclusions

Proper fault prognostic methods of modeling the degradation path of micro-switches are urgent for the RUL estimation and appropriate period maintenance decision in MEMS devices. This paper proposes a novel effective method as a linear degradation model based on an inverse Kalman filter for evaluating the approximately accurate RUL of the micro-switches. Firstly, Bayesian posterior estimation and EM algorithm were used to estimate the stochastic parameters. Then, an inverse Kalman filter was delivered to solve the errors of the initial parameters, and the STF method was proposed on the basis of the Bayesian updating in order to improve the accuracy of estimating the nonlinear data. Next, the effectiveness of the proposed approach was validated on experimental data relating to micro-switches of the rail vehicles. Finally, a series of comprehensive and persuasive comparison experiments proved to illustrate the effectiveness of the method with an inverse Kalman filter. In future work, the proposed method in this paper may contribute to the analysis of prediction methods of other MEMS devices. And it is inspired by the extended Kalman filter (EKF), which will play a positive role in the RUL prediction of nonlinear stochastic processes.

**Author Contributions:** In this work, B.Z., Y.S. (Yubo Shao) and Y.S. (Yuankun Sui) conceived and desgned the experiments; Y.S. (Yuankun Sui) performed the experiments; Y.S. (Yubo Shao) and Z.S. analyzed the data; Z.C. contributed analysis tools; Y.S. (Yubo Shao) wrote the paper.

**Funding:** This work was supported by the National Natural Science Fund (NSFC) under Grant 61751304. It was also supported by Jilin Province Science and Technology Department Key Technology R&D Project (Special Support) EMU Fault Prediction and Health Management System R&D 20180201125GX. In addition, it was supported by Changchun Science and Technology Plan Project 17CX001, Changchun Rail Vehicle Electric Control System Technology R&D Center. As well it was supported by Changchun Science and Technology Project 18SS011.

**Conflicts of Interest:** The authors declare no conflict of interest.

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
