# Peer review of "A Stochastic Deterioration Process Based Approach for Micro Switches Remaining Useful Life Estimation"

_applsci, doi:10.3390/app9030613_

Reviewer 1 Report

The authors consider real time estimation RUL using a two stage procedure. The first an empirical Bayes estimation of the parameters, the second an `inverse' Kalman filter initialization with STF.

There are some issues at the beginning of the article. The model used in section 2.1; eq (2.1) looks wrong to me; if you literally substitute the time increment $t_k-t_{k-1}$ into eq (1) (i.e. the drifted Brownian motion with scale sigma), then clearly eq (2) is not correct. So one must clarify this.

The `EM algorithm'. Well here one of course knows pointwise the marginal likelihood and in principle many methods can be used such as Newton Raphson etc (do we even need a numerical method?). Perhaps EM is the easiest, although whatever method they use, it is so trivial that we don't need any details. The authors have to clarify that in fact what they are doing is an empirical Bayes type method, although it is strange that $\sigma$ is also estimated this way. The model is so simple, why not put a prior on $\sigma$? It would be straightforward to get a posterior on $\sigma$. This opening is important to be properly written.

There are several grammatical issues, including a persistent incorrect usage of `the'.

Author Response

Dear Reviewers:

Thank you for your letter and for the reviewers’ comments concerning our manuscript entitled “A Stochastic Deterioration Process Based Approach for Micro switches Remaining Useful Life Estimation” (applsci-424353).Those comments are all valuable and very helpful for revising and improving our paper, as well as the important guiding significance to our researches. We have studied comments carefully and have made correction which we hope meet with approval. Revised portion are improved with “Track Changes” function of Microsoft Word in the paper. The main corrections in the paper and the responds to the reviewer’s comments are illustrated in “Response 1”.

We appreciate for Reviewers’ warm work earnestly, and hope that the correction will meet with approval. Once again, thank you very much for your comments and suggestions.

Yours Sincerely,

Yubo Shao

Reviewer 2 Report

This paper generally presents the prognostic method for remaining useful life estimation. The following are suggestions to improve the quality of the paper prior to publication:
1. The paper used Kalman filter method for micro switches remaining useful life estimation. There are a number of prognostic methods are presented in literature, the reason why Kalman filter is selected in the paper should be described comprehensively.
   I suggests the Authors include some of recent review papers on Prognostics methods.
   For example:
   1. Machinery health prognostics: A systematic review from data acquisition to RUL prediction
   2. Parsimonious Network Based on a Fuzzy Inference System (PANFIS) for Time Series Feature Prediction of Low Speed Slew Bearing Prognosis
2. The fontsize inside the block of Figure 1 is too small and a bit difficult to see.
3. Some sentences are need to be improved:
   For example in line 327: "The test rig is same with Zhang et.al. [30]".
   Probably it can be improved as follow: "The test rig used in this experiment is identical with the one used by Zhang et.al. [30]".
4. The caption of Figures 4-10 are too small and a bit difficult to see. Please improve the figure presentation to better visualization.
5. The results of proposed method is compared with the result of Wang [19] as presented in Figures 7-10. Is Wang [19] also estimate the RUL of micro switches?
6. Since the test rig is identical with Zhang et.al. [30] and some parameters are also similar (line 333), why the Authors did not compare the results of proposed method with the result of Zhang [30]?

Author Response

Dear Reviewers:

Thank you for your letter and for the reviewers’ comments concerning our manuscript entitled “A Stochastic Deterioration Process Based Approach for Micro switches Remaining Useful Life Estimation” (applsci-424353).Those comments are all valuable and very helpful for revising and improving our paper, as well as the important guiding significance to our researches. We have studied comments carefully and have made correction which we hope meet with approval. Revised portion are improved with “Track Changes” function of Microsoft Word in the paper. The main corrections in the paper and the responds to the reviewer’s comments are illustrated in “Response 2”.

We appreciate for Reviewers’ warm work earnestly, and hope that the correction will meet with approval. Once again, thank you very much for your comments and suggestions.                           

Yours Sincerely,

Yubo Shao

Reviewer 3 Report

The paper is very good and deserves publication.

The paper considers not only theoretical aspects but validates these aspects using experimental measurements.

Please clarify better the difference between the traditional Kalman Filter and the proposed one.

In particular, please explain the concept of the "flipped data" mentioned in Remark 1. This is important because a question arises:

How to apply online this flip data?

All figures of the paper are too small or with too small characters and must be improved such that the reader can profit from it.

Concerning the literature the authors should read the following papers and the literature therein to be inspired for future works.

In the following paper observers are developed to be applied in mechanical systems and a comparison is done with the Kalman Filter considering the  same application.

This paper could be useful to contestualize the General issue of this Topic.

Mercorelli, P. A Motion-Sensorless Control for Intake Valves in Combustion Engines (2017) IEEE Transactions on Industrial Electronics, 64 (4), art. no. 7534788, pp. 3402-3412.

Author Response

Dear Reviewers:

Thank you for your letter and for the reviewers’ comments concerning our manuscript entitled “A Stochastic Deterioration Process Based Approach for Micro switches Remaining Useful Life Estimation” (applsci-424353).Those comments are all valuable and very helpful for revising and improving our paper, as well as the important guiding significance to our researches. We have studied comments carefully and have made correction which we hope meet with approval. Revised portion are improved with “Track Changes” function of Microsoft Word in the paper. The main corrections in the paper and the responds to the reviewer’s comments are illustrated in “Response 3”.

We appreciate for Reviewers’ warm work earnestly, and hope that the correction will meet with approval. Once again, thank you very much for your comments and suggestions.                                        

Yours Sincerely,

Yubo Shao

Round  2

Reviewer 1 Report

The paper is acceptable in it's present form.

Reviewer 2 Report

Dear Authors,

Thank you for providing the revised version of the paper. I have no further comments.

Kind regards,

- Reviewer -